# A multi-population phenome-wide association study of genetically-predicted height in the Million Veteran Program

Sridharan Raghavan[1,2☯*], Jie Huang[3☯], Catherine Tcheandjieu[4,5], Jennifer E. Huffman[6], Elizabeth Litkowski[1], Chang Liu[7,8], Yuk-Lam A. Ho[6], Haley Hunter-Zinck[6], Hongyu Zhao[9,10], Eirini Marouli[11], Kari E. North[12], the VA Million Veteran Program, Ethan Lange[2], Leslie A. Lange[2], Benjamin F. Voight[13,14,15,16], J. Michael Gaziano[6,17], Saiju Pyarajan[6,17], Elizabeth R. Hauser[18,19], Philip S. Tsao[4,5], Peter W. F. Wilson[7,20], Kyong-Mi Chang[13,21], Kelly Cho[6,17], Christopher J. O'Donnell[6,17], Yan V. Sun[7,8], Themistocles L. Assimes[4,5]

1 Medicine Service, Veterans Affairs Eastern Colorado Health Care System, Aurora, Colorado, United States of America, 2 Department of Medicine, University of Colorado Anschutz Medical Campus, Aurora, Colorado, United States of America, 3 School of Public Health and Emergency Management, Southern University of Science and Technology, Shenzhen, Guangdong, China, 4 Veterans Affairs Palo Alto Health Care System, Palo Alto, California, United States of America, 5 Division of Cardiovascular Medicine, Department of Medicine, Stanford University School of Medicine, Stanford, California, United States of America, 6 Veterans Affairs Boston Healthcare System, Boston, Massachusetts, United States of America, 7 Atlanta Veterans Affairs Medical Center, Atlanta, Georgia, United States of America, 8 Department of Epidemiology, Emory University Rollins School of Public Health, Atlanta, Georgia, United States of America, 9 Veterans Affairs Connecticut Healthcare System, West Haven, Connecticut, United States of America, 10 Department of Biostatistics, Yale University School of Public Health, New Haven, Connecticut, United States of America, 11 William Harvey Research Institute, Barts and The London School of Medicine and Dentistry, Queen Mary University of London, London, United Kingdom, 12 Department of Epidemiology, Gillings School of Public Health, University of North Carolina, Chapel Hill, North Carolina, United States of America, 13 Corporal Michael J. Crescenz Veterans Affairs Medical Center, Philadelphia, Pennsylvania, United States of America, 14 Department of Genetics, University of Pennsylvania Perelman School of Medicine, Philadelphia, Pennsylvania, United States of America, 15 Department of Systems Pharmacology and Translational Therapeutics, University of Pennsylvania Perelman School of Medicine, Philadelphia, Pennsylvania, United States of America, 16 Institute of Translational Medicine, University of Pennsylvania Perelman School of Medicine, Philadelphia, Pennsylvania, United States of America, 17 Department of Medicine, Brigham and Women's Hospital, Harvard Medical School, Boston, Massachusetts, United States of America, 18 Cooperative Studies Program Epidemiology Center- Durham, Durham Veterans Affairs Health Care System, Durham, North Carolina, United States of America, 19 Department of Biostatistics and Bioinformatics, Duke University Medical Center, Durham, North Carolina, United States of America, 20 Division of Cardiology, Emory University School of Medicine, Atlanta, Georgia, United States of America, 21 Department of Medicine, University of Pennsylvania Perelman School of Medicine, Philadelphia, Pennsylvania, United States of America

☯ These authors contributed equally to this work.
* Sridharan.raghavan@cuanschutz.edu

**Data Availability Statement:** Ancestry-stratified results reported in the manuscript for the PheWAS of height and genetically-predicted height are available in the Supporting Information (S11–S14

## Abstract

### Background

Height has been associated with many clinical traits but whether such associations are causal versus secondary to confounding remains unclear in many cases. To systematically examine this question, we performed a Mendelian Randomization-Phenome-wide association study (MR-PheWAS) using clinical and genetic data from a national healthcare system biobank.

Tables). Summary statistics of GWAS are available through the Database of Phenotypes and Genotypes (dbGaP) accession number phs001672 (https://www.ncbi.nlm.nih.gov/projects/gap/cgi-bin/study.cgi?study_id=phs001672). Data access is restricted in that dbGaP requires an application requesting data per their standard procedures.

**Funding:** This work was supported by funding from the US Department of Veterans Affairs (https://www.research.va.gov/) MVP Program awards MVP001 I01-BX004821 (YH, KC, PWFW) and MVP003/028 I01-BX003362 (CT, PST, KMC, TLA); by the US Department of Veterans Affairs (https://www.research.va.gov/) award IK2-CX001907 (SR); by funds from the Boettcher Foundation's Webb-Waring Biomedical Research Program (https://boettcherfoundation.org/webb-waring-biomedical-research/) (SR); by the US National Institutes of Health, National Institute for Diabetes, Digestive, and Kidney Diseases (https://www.niddk.nih.gov/research-funding) awards R01DK122503 (KEN), R01DK101855 (KEN), DK101478 (BFV), and DK126194 (BFV); by the US National Institutes of Health, National Human Genome Research Institute (https://www.genome.gov/research-funding) awards R01HG010297 (KEN) and R01HG009974 (KEN); by the US National Institutes of Health, National Heart, Lung, and Blood Institute (https://www.nhlbi.nih.gov/grants-and-training) awards R01HL142302 (KEN) and R01HL143885 (KEN); and by a Linda Pechenick Montague Investigator award (BFV). The funders had no role in study design, data collection and analysis, decision to publish, or preparation of the manuscript.

**Competing interests:** I have read the journal's policy and the authors of this manuscript have the following competing interests: CJO is a full-time employee of Novartis Institutes of Biomedical Research. The remaining authors have declared that no competing interests exist.

## Methods and findings

Analyses were performed using data from the US Veterans Affairs (VA) Million Veteran Program in non-Hispanic White (EA, n = 222,300) and non-Hispanic Black (AA, n = 58,151) adults in the US. We estimated height genetic risk based on 3290 height-associated variants from a recent European-ancestry genome-wide meta-analysis. We compared associations of measured and genetically-predicted height with phenome-wide traits derived from the VA electronic health record, adjusting for age, sex, and genetic principal components. We found 345 clinical traits associated with measured height in EA and an additional 17 in AA. Of these, 127 were associated with genetically-predicted height at phenome-wide significance in EA and 2 in AA. These associations were largely independent from body mass index. We confirmed several previously described MR associations between height and cardiovascular disease traits such as hypertension, hyperlipidemia, coronary heart disease (CHD), and atrial fibrillation, and further uncovered MR associations with venous circulatory disorders and peripheral neuropathy in the presence and absence of diabetes. As a number of traits associated with genetically-predicted height frequently co-occur with CHD, we evaluated effect modification by CHD status of genetically-predicted height associations with risk factors for and complications of CHD. We found modification of effects of MR associations by CHD status for atrial fibrillation/flutter but not for hypertension, hyperlipidemia, or venous circulatory disorders.

## Conclusions

We conclude that height may be an unrecognized but biologically plausible risk factor for several common conditions in adults. However, more studies are needed to reliably exclude horizontal pleiotropy as a driving force behind at least some of the MR associations observed in this study.

### Author summary

Adult height has been associated with several clinical traits, for example with increased risk of atrial fibrillation and with decreased risk of cardiovascular disease. Using data from the VA Million Veteran Program that includes genetic data linked to clinical records in >200,000 non-Hispanic White adults and >50,000 non-Hispanic Black adults, we examined associations of measured height and genetically-predicted height with clinical traits phenome-wide. By comparing associations of traits with measured and with genetically-predicted height, we aimed to discriminate between potentially causal associations (those associated with genetically-predicted height) from associations that may be confounded by environmental exposures over the life course (those associated with measured height but not with genetically-predicted height). Of approximately 350 traits associated with measured height, we found 127 associated with genetically-predicted height in non-Hispanic White individuals. While only 2 were also statistically significant in non-Hispanic Black individuals, we found evidence for consistent directions of effect for associations of traits with genetically-predicted height in non-Hispanic Black and White individuals. We conclude that height may be an unrecognized non-modifiable risk factor for several common conditions in adults.

## Introduction

Height is not typically considered a disease risk factor but has nevertheless been associated with numerous diseases [1–5]. Such epidemiologic associations of height with disease end-points are susceptible to confounding as adult height is also influenced by environmental factors, including nutrition, socioeconomic status, and demographic factors [1,6–9]. The high heritability of height coupled with recent advances in understanding its genetic basis [10–12] now make it possible to use genetic tools to elucidate pathophysiologic relationships between height and clinical traits.

Mendelian randomization (MR) is an instrumental variable approach that utilizes genetic instruments for exposures of interest under the assumption that genotype is less susceptible to confounding than measured exposures [13]. Indeed, MR has recently been used to address unmeasured confounding and to estimate causal effects of height on several candidate traits of interest, including coronary heart disease (CHD), lipid levels, atrial fibrillation, and certain cancers [6,14–20]. The largest of the MR studies examined height associations with 50 traits using data from the UK Biobank and 691 height-associated genetic variants from a European-ancestry genome-wide association study (GWAS) meta-analysis [11,15]. Twelve of the 50 traits studied had genetic evidence supporting an association with height: risk-lowering associations of taller stature with heart disease, hypertension, diaphragmatic hernia, and gastroesophageal reflux disease, and risk-increasing associations of height with atrial fibrillation, venous thromboembolic events, hip fracture, intervertebral disc disease, vasculitis, and cancer (all-cause as well as breast and colorectal cancers specifically)[15]. Thus, the prior work expands the genetic evidence supporting height associations beyond cardiovascular disease and cancer to gastrointestinal, musculoskeletal, and rheumatologic diseases.

While the prior MR studies of height have tested hypotheses based on previously described epidemiologic associations, MR methods have also been combined with phenome-wide association studies (MR-PheWAS) to identify novel or hypothesis-generating associations [21]. For example, using a genetic instrument for body mass index (BMI) and phenotype data from the UK Biobank, a recent study evaluated genetic evidence for associations of BMI with nearly 20,000 independent traits and identified novel associations with psychiatric traits related to nervousness [22]. A similar comprehensive or phenome-wide evaluation of clinical traits associated with measured and genetically-predicted height could elucidate the full scope of diseases associated with height as a risk or protective factor. To that end, we performed an MR-PheWAS of height in the multi-population US Department of Veterans Affairs (VA) Million Veteran Program (MVP).

## Results

### Study participants

Among 323,793 MVP participants with both genetic data and height measurements, there were 73% of non-Hispanic White (EA) (n = 235,398), 20% of non-Hispanic Black (AA) (n = 63,898), and 7.6% of Hispanic (HA) (n = 24,497) race/ethnicity. The MVP participants were predominantly men (91.6%) with a mean height of 176 cm and mean BMI of 30.1 kg/m$^2$ (**Table 1**). AA and EA participants were similar with regard to sex, BMI, and height, but AA participants were younger on average (57.7 years versus 64.2 years). HA participants were younger (mean age 55.7 years) and shorter (172 cm) than AA and EA counterparts (**Table 1**). Of these, 280,451 individuals (222,300 EA and 58,151 AA) were included in the PheWAS analysis based on availability of electronic health record phenotypes needed for PheWAS. We did not include HA in the PheWAS due to limited sample size relative to EA and AA but the HA

**Table 1. Study participant characteristics.**

| | All participants<br>N = 323,793 | Non-Hispanic White<br>N = 235,398 | Non-Hispanic Black<br>N = 63,898 | Hispanic-American<br>N = 24,497 |
|---|---|---|---|---|
| Age (years), mean (SD) | 62.2 (13.5) | 64.2 (13.2) | 57.7 (11.9) | 55.7 (15.1) |
| Sex, n (%) | | | | |
| *Male* | 297,470 (91.9) | 219,177 (93.1) | 55,837 (87.4) | 22,456 (91.7) |
| *Female* | 26,323 (8.1) | 16,221 (6.9) | 8,061 (12.6) | 2,041 (8.3) |
| Body mass index (kg/m$^2$), mean (SD) | 30.1 (5.9) | 30.1 (5.9) | 30.3 (6.2) | 30.7 (5.6) |
| Height (cm), mean (SD) | 176 (7.8) | 177 (7.5) | 177 (8.3) | 172 (7.4) |

sample is included in an ongoing multi-biobank PheWAS effort through the GIANT consortium with a larger sample size [23]. The electronic health records of EA and AA individuals reflected similar numbers of phecodes: 46±33 phecodes in EA and 48±32 phecodes in AA (mean±SD).

## Transferability of genetic associations with height to the MVP population

The GRS based on 3,290 SNPs independently associated with height in a prior European-ancestry meta-analysis that did not include MVP explained 18% of height variation in EA and 4.8% of height variation in AA in MVP. Of the 3,290 variants examined, 89% had concordant direction of effect between the discovery study and MVP EA ($p < 2 \times 10^{-16}$) and 72% had concordant direction of effect between the discovery study and MVP AA ($p < 2 \times 10^{-16}$). The standardized effect sizes for associations between the 3,290 SNPs and height were well-correlated between the discovery study and MVP EA and AA with a best-fit line slope of 0.87 [95% CI 0.84, 0.90] in EA and of 0.78 [0.73, 0.82] in AA (**S1 Fig**).

## Association of height with clinical traits

A total of 345 traits were associated with measured height at phenome-wide significance ($p < 3.6 \times 10^{-5}$) among EA individuals, 133 of which were phenome-wide significant ($p < 5.0 \times 10^{-5}$) in AA individuals as well (**Fig 1A**). An additional 17 traits were phenome-wide significant in AA individuals but not in EA individuals (**S1 and S2 Tables**). The standardized effect estimates (Z') for height-trait associations that reached phenome-wide significance in at least one race/ethnicity group were well correlated (r = 0.83, $p = 1.4 \times 10^{-91}$) between EA and AA individuals. The regression line of best fit (EA as Y ~ AA as X) between the standardized effect estimates (Z') for height-trait associations that reached phenome-wide significance in at least one race/ethnicity group had a slope of 0.94 [95% CI 0.88, 0.99] (**Fig 1A**, correlation plot of unstandardized odds ratios [OR] in EA and AA). Out of 362 height-trait associations that were phenome-wide significant in at least one race/ethnicity group (212 in EA only, 133 in both AA and EA, and 17 in AA alone), 335 had concordant directions of effect between the two race/ethnicity groups ($p < 2 \times 10^{-16}$).

## Association of genetically-predicted height with clinical traits

A total of 142 traits were associated with genetically-predicted height at phenome-wide significance among EA individuals, 2 of which were phenome-wide significant in AA individuals as well, and another 46 of which were nominally significant ($p < 0.05$) in AA individuals (**Fig 1B** and **S3 and S4 Tables**). The standardized effect estimates (Z') for phenome-wide significant associations of clinical traits with genetically-predicted height were reasonably correlated between EA and AA individuals (r = 0.83, p = $2.0 \times 10^{-37}$) and improved when limited to traits

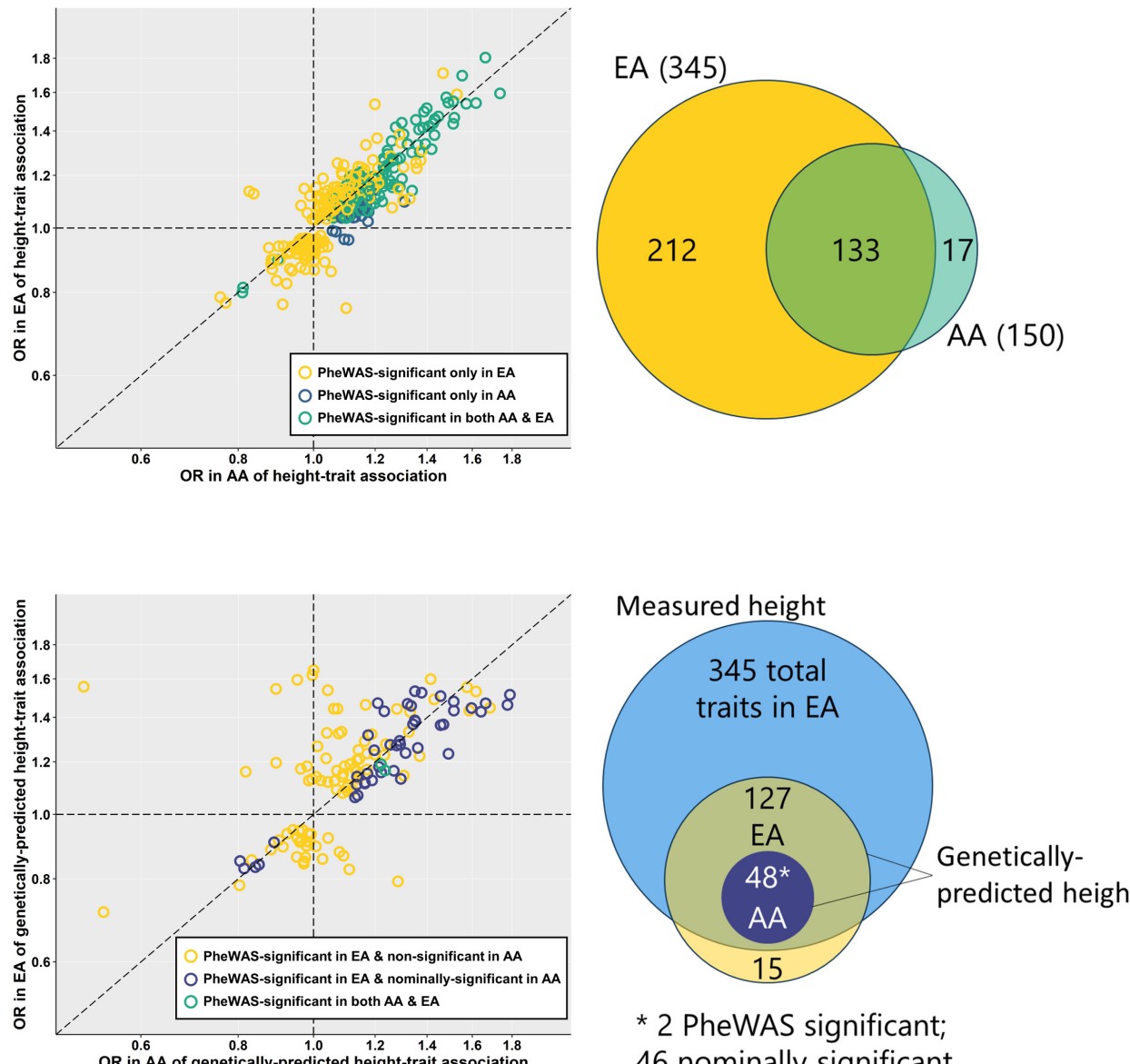

**Fig 1. Comparison of number of associations and effect sizes of measured height (A) and genetically-predicted height (B) between non-Hispanic White and non-Hispanic Black individuals.** Associations of measured and genetically-predicted height with phecodes represented as odds ratios (OR) in non-Hispanic Black (AA) and non-Hispanic White (EA) MVP participants. Whether associations exceeded phenome-wide significance threshold in either or both race/ethnicity groups indicated by color. Venn diagrams providing pictorial representation of the same comparisons shown to the right of each plot.

that were at least nominally significant in AA individuals (r = 0.92, p = 1.2×10⁻²⁰). The regression line of best fit comparing standardized effect estimates for phenome-wide significant associations of clinical traits with genetically-predicted height in EA and AA had a slope of 1.85 [95% CI 1.67, 2.02] (with EA as Y ~ AA as X) and improved to 1.80 [95% CI 1.64, 1.96] when limited to traits that were at least nominally significant in AA individuals (**Fig 1B**, correlation plot of unstandardized OR in EA and AA). That the slope of the best fit line was >1 implies larger effect sizes in EA than in AA but such a quantitative interpretation is limited by a number of factors: differences in sample size, winner's curse, and differences in the strength

of the GRS as a genetic instrument in the two populations. An appropriately cautious interpretation would be that the regression slope and plots comparing effect sizes are consistent with concordant directions of effect for trait associations with genetically-predicted height between EA and AA individuals in MVP. Though only 48 out of 142 traits associated at phenome-wide significance with genetically-predicted height in EA were at least nominally significant in AA, 124 out of those 142 traits had the same direction of effect in EA and AA for association with genetically-predicted height ($p = 4 \times 10^{-16}$).

A total of 127 traits among EA individuals were associated with genetically-predicted height and measured height at phenome-wide significance (**Fig 2** and **S3 Table**). Thus, we found genetic evidence supporting associations with height for 37% (127/345) of the traits that were

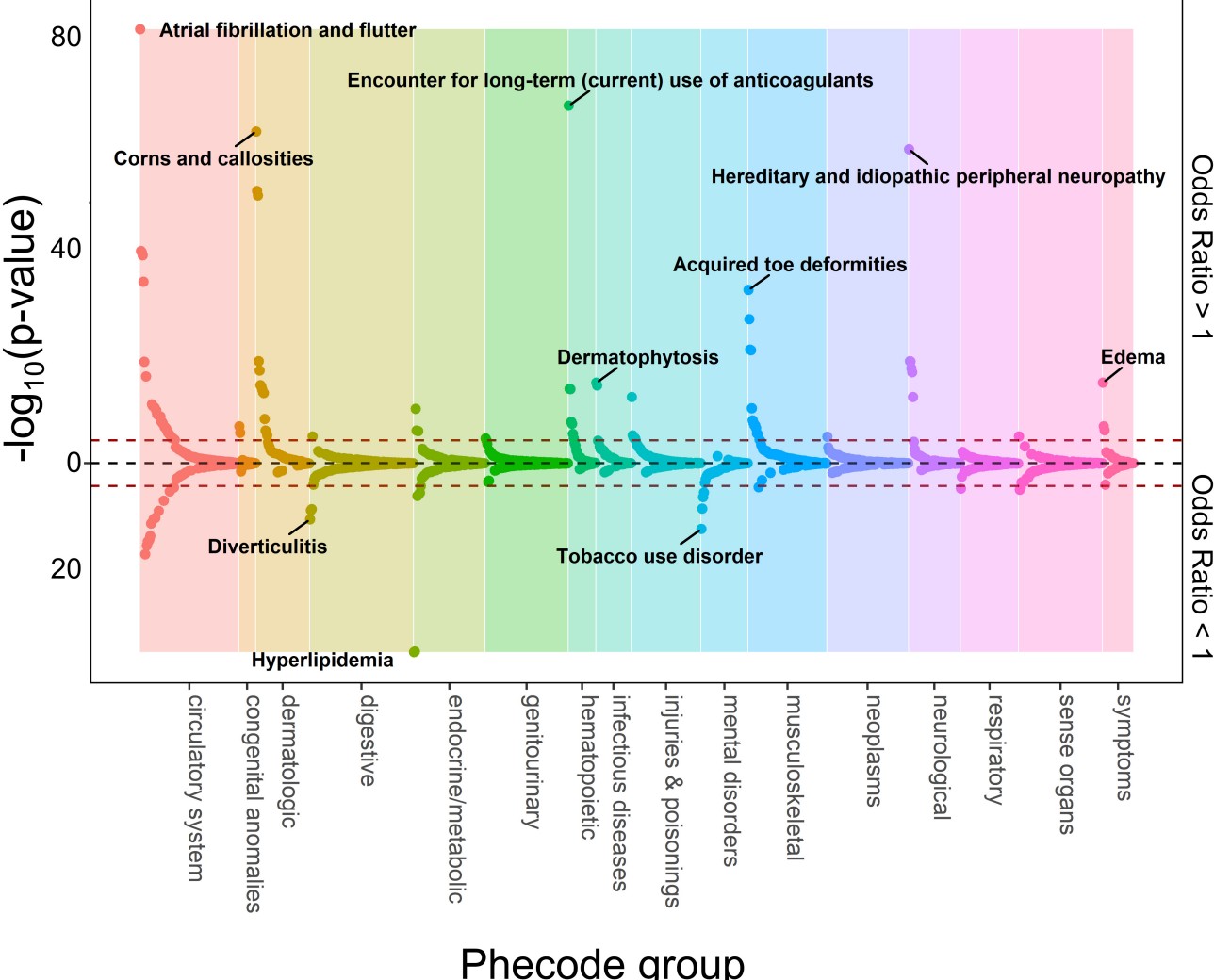

**Fig 2. Phenome-wide associations with genetically predicted height in non-Hispanic White individuals.** Plot of phecodes versus -log₁₀(p-value) for association with genetically-predicted height in non-Hispanic White participants in MVP. Phecodes were limited to single decimal place for clarity (e.g., 427.2 for atrial fibrillation or flutter is shown but 427.21 for atrial fibrillation is not). Associations with a negative beta coefficient (i.e., odds ratio < 1) are plotted below the x-axis, and those with a positive beta coefficient (i.e., odds ratio > 1) are plotted above the x-axis. Red dotted lines indicate race/ethnicity-specific phenome-wide significance thresholds ($p < 3.6E\text{-}5$ for non-Hispanic White). The top association (lowest p-value) within each phecode group is labeled.

associated with measured height. In AA individuals, 2 traits (acquired foot deformities [phecode 735] and dermatophytosis of nail [phecode 110.11]) were associated with genetically-predicted height and measured height at phenome-wide significance (**Fig 3** and **S4 Table**). An additional 46 traits had nominally-significant associations with genetically-predicted height and phenome-wide significant associations with measured height in AA individuals. Of 11 traits with directionally consistent associations with measured height and genetically-predicted height previously reported in the UK Biobank [15], 9 had analogous phecodes in MVP of which 7 replicated with the same direction of effect for the association with genetically-predicted height with at least nominal significance in MVP EA individuals (**S5 Table**).

BMI is proportional to the inverse of height-squared, and we found that obesity was inversely associated with genetically predicted height (Odds Ratio [OR] 0.94 per SD increase

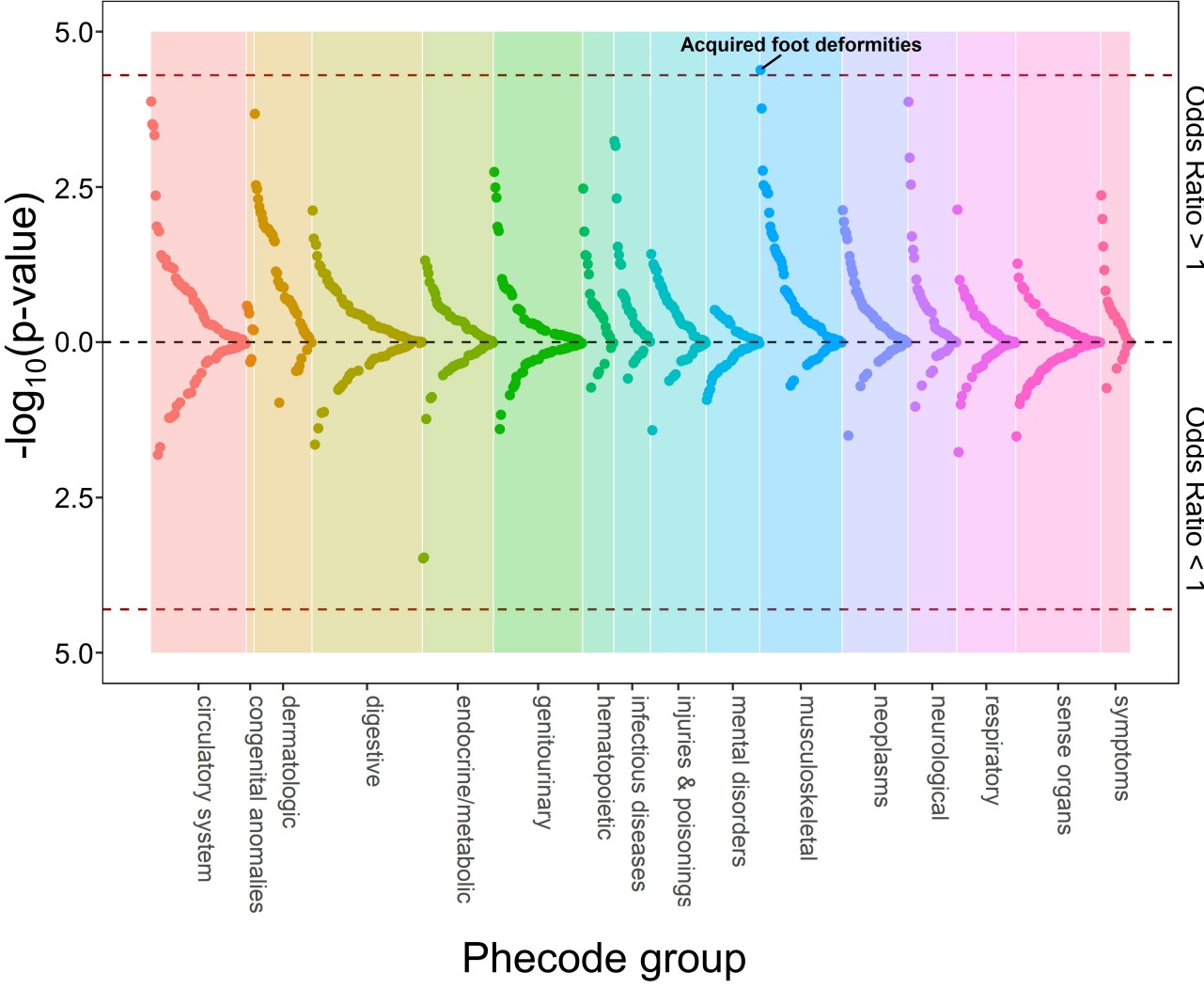

**Fig 3. Phenome-wide associations with genetically predicted height in non-Hispanic Black individuals.** Plot of phecodes versus -$\log_{10}$(p-value) for association with genetically-predicted height in non-Hispanic Black participants in MVP. Phecodes were limited to single decimal place for clarity (e.g., 427.2 for atrial fibrillation or flutter is shown but 427.21 for atrial fibrillation is not). Associations with a negative beta coefficient (i.e., odds ratio < 1) are plotted below the x-axis, and those with a positive beta coefficient (i.e., odds ratio > 1) are plotted above the x-axis. Red dotted lines indicate race/ethnicity-specific phenome-wide significance thresholds ($p < 5.0E-5$ for non-Hispanic Black). The top association (lowest p-value) within each phecode group is labeled.

in height, $p = 5.5 \times 10^{-7}$). To address potential confounding by BMI, we tested whether associations of genetically-predicted height with clinical traits were sensitive to inclusion of BMI as a covariate in the MR-PheWAS analysis. Inclusion of BMI as a covariate did not substantially impact the beta coefficients for traits associated at phenome-wide significance in EA individuals and at nominal significance in AA individuals (S2 Fig).

Meta-analyzing genetically-predicted height associations from EA and AA individuals identified 141 total traits associated with genetically-predicted height at phenome-wide significance ($p < 3.6 \times 10^{-5}$). Of these, five were phenome-wide significant only after meta-analysis and not in either race/ethnicity group alone (S6 Table). Table 2 shows the top ten and closely related phecodes associated with genetically-predicted height in EA individuals that achieve at least nominal significance in AA individuals, all of which were phenome-wide significant in the meta-analysis of EA and AA associations. Circulatory system was the most frequent phecode group/system among the top ten, consistent with prior studies of height-associated conditions [2–5,14–16,18–20].

We repeated the MR-PheWAS after stratifying by sex within EA and AA individuals. In EA individuals, there were 1308 phecodes with ≥200 cases in men (significance $p < 3.8 \times 10^{-5}$) and 598 phecodes with ≥200 cases in women (significance $p < 8.4 \times 10^{-5}$). There were 135 and 6 phenome-wide significant traits associated with genetically-predicted height in EA men and women, respectively. Of the 6 phecodes associated with genetically-predicted height in EA women at phenome-wide significance, 2 were significant only in women (phecode 495, Asthma; phecode 351, Other peripheral nerve disorders), and 4 were also phenome-wide significant in men (S7 Table). The standardized effect estimates (Z') for phenome-wide significant associations of clinical traits with genetically-predicted height were reasonably correlated between EA men and women (r = 0.86, $p = 5.4 \times 10^{-40}$; best fit line slope 0.97 [95% CI 0.87, 1.08] with men as Y ~ women as X). In AA individuals, there were 923 and 348 phecodes with ≥200 cases in men and women, respectively, corresponding to significance thresholds of $p < 5.4 \times 10^{-5}$ in men and $p < 1.4 \times 10^{-4}$ in women. We did not identify any phecodes associated with genetically-predicted height at phenome-wide significance in stratified analyses of AA men or AA women.

## Effect of CHD status on genetically-predicted height associations with CHD risk factors and other circulatory system disorders

Genetically-predicted height has previously been associated with CHD and several CHD risk factors [15,18], and we reproduced several of these associations (S3 Table). Given that CHD risk factors and CHD are correlated in our clinical data, we examined associations of genetically-predicted height with hyperlipidemia and hypertension stratifying by CHD status. That is, by stratifying by CHD status, we attempted to examine independence of genetically-predicted height associations with CHD and with hyperlipidemia, hypertension, atrial fibrillation, and venous circulatory disorders. Genetically-predicted height was inversely associated with hyperlipidemia and hypertension in both individuals without and with CHD (heterogeneity $p > 0.05$ for both; Fig 4). Atrial fibrillation or flutter is another cardiovascular condition associated with height and genetically-predicted height [5,16,19]. CHD is also a risk factor for atrial fibrillation, but the associations of genetically-predicted height with CHD and atrial fibrillation are in opposing directions. We found that genetically-predicted height was associated with a higher odds ratio for atrial fibrillation or flutter in individuals without CHD (OR 1.51 [95% CI 1.43, 1.59] per SD increase in height) than in those with CHD (OR 1.39 [95% CI 1.32, 1.46] per SD increase in height; heterogeneity $p = 0.03$; Fig 4). Conversely, we did not observe heterogeneity in the associations of genetically-predicted height with varicose veins of the lower extremity or

**Table 2. Top phenome-wide associations and closely related traits with genetically predicted height.**

| Phecode | Description | Group/System | Meta-analysis | | Non-Hispanic White | | | | | Non-Hispanic Black | | | | |
|---|---|---|---|---|---|---|---|---|---|---|---|---|---|---|
| | | | OR | *p* | OR | *p* | N Cases | N controls | N total | OR | *p* | N Cases | N Controls | N Total |
| 427.2 | Atrial fibrillation and flutter | circulatory system | 1.38 | 9.1E-87 | 1.38 | 5.7E-84 | 31478 | 117588 | 149066 | 1.35 | 3.3E-04 | 3557 | 32339 | 35896 |
| 427.21 | Atrial fibrillation | circulatory system | 1.38 | 1.1E-85 | 1.39 | 4.2E-83 | 30266 | 117588 | 147854 | 1.35 | 5.3E-04 | 3314 | 32339 | 35653 |
| 427.22 | Atrial flutter | circulatory system | 1.37 | 7.8E-25 | 1.36 | 2.4E-23 | 6854 | 117588 | 124442 | 1.46 | 8.3E-03 | 1106 | 32339 | 33445 |
| 700 | Corns and callosities | dermatologic | 1.44 | 1.7E-64 | 1.47 | 2.2E-64 | 12823 | 165389 | 178212 | 1.21 | 2.9E-03 | 6056 | 41519 | 47575 |
| 356 | Hereditary and idiopathic peripheral neuropathy | neurological | 1.37 | 3.9E-64 | 1.37 | 5.8E-61 | 18742 | 184942 | 203684 | 1.34 | 1.3E-04 | 3880 | 48952 | 52832 |
| 337 | Disorders of the autonomic nervous system | neurological | 1.44 | 4.4E-22 | 1.43 | 2.8E-20 | 4183 | 170336 | 174519 | 1.64 | 2.9E-03 | 779 | 45621 | 46400 |
| 337.1 | Peripheral autonomic neuropathy | neurological | 1.48 | 2.5E-22 | 1.46 | 3.1E-20 | 3666 | 170336 | 174002 | 1.78 | 1.1E-03 | 700 | 45621 | 46321 |
| 357 | Inflammatory and toxic neuropathy | neurological | 1.48 | 6.3E-20 | 1.48 | 6.4E-19 | 3209 | 184942 | 188151 | 1.51 | 3.2E-02 | 567 | 48952 | 49519 |
| 707.2 | Chronic ulcer of leg or foot | dermatologic | 1.52 | 2.1E-54 | 1.53 | 5.7E-53 | 8181 | 203450 | 211631 | 1.35 | 4.9E-03 | 1955 | 53715 | 55670 |
| 707 | Chronic ulcer of skin | dermatologic | 1.42 | 1.0E-52 | 1.43 | 4.2E-52 | 11694 | 203450 | 215144 | 1.23 | 2.2E-02 | 2716 | 53715 | 56431 |
| 735.21 | Hammer toe (acquired) | musculoskeletal | 1.51 | 3.0E-55 | 1.52 | 1.6E-52 | 8466 | 161389 | 169855 | 1.38 | 2.1E-04 | 3095 | 37585 | 40680 |
| 735.2 | Acquired toe deformities | musculoskeletal | 1.29 | 4.7E-37 | 1.29 | 5.8E-34 | 15251 | 161389 | 176640 | 1.29 | 1.7E-04 | 5317 | 37585 | 42902 |
| 735 | Acquired foot deformities | musculoskeletal | 1.19 | 5.3E-32 | 1.19 | 2.2E-28 | 29185 | 161389 | 190574 | 1.22 | 4.1E-05 | 11876 | 37585 | 49461 |
| 454.1 | Varicose veins of lower extremity | circulatory system | 1.48 | 2.0E-44 | 1.47 | 2.0E-41 | 8031 | 126503 | 134534 | 1.66 | 1.3E-04 | 1256 | 32810 | 34066 |
| 454 | Varicose veins | circulatory system | 1.44 | 2.9E-43 | 1.43 | 1.2E-40 | 9084 | 126503 | 135587 | 1.51 | 4.6E-04 | 1599 | 32810 | 34409 |
| 456 | Chronic venous insufficiency [CVI] | circulatory system | 1.37 | 2.8E-38 | 1.37 | 1.6E-35 | 10740 | 126503 | 137243 | 1.47 | 3.1E-04 | 2053 | 32810 | 34863 |
| 272.1 | Hyperlipidemia | endocrine/ metabolic | 0.84 | 9.1E-40 | 0.84 | 5.7E-37 | 165197 | 44435 | 209632 | 0.85 | 3.3E-04 | 37725 | 16430 | 54155 |
| 272 | Disorders of lipoid metabolism | endocrine/ metabolic | 0.84 | 1.2E-39 | 0.84 | 7.5E-37 | 165335 | 44435 | 209770 | 0.85 | 3.4E-04 | 37776 | 16430 | 54206 |
| 272.13 | Mixed hyperlipidemia | endocrine/ metabolic | 0.83 | 3.5E-25 | 0.83 | 1.0E-23 | 35369 | 44435 | 79804 | 0.84 | 9.8E-03 | 7552 | 16430 | 23982 |
| 272.11 | Hypercholesterolemia | endocrine/ metabolic | 0.83 | 3.3E-25 | 0.83 | 5.2E-23 | 33451 | 44435 | 77886 | 0.81 | 1.5E-03 | 8713 | 16430 | 25143 |
| 110.11 | Dermatophytosis of nail | infectious diseases | 1.17 | 2.4E-27 | 1.16 | 1.1E-23 | 33539 | 146509 | 180048 | 1.24 | 2.3E-05 | 12663 | 31035 | 43698 |
| 710.1 | Osteomyelitis | musculoskeletal | 1.46 | 2.7E-23 | 1.47 | 1.7E-22 | 4032 | 165278 | 169310 | 1.32 | 4.1E-02 | 1184 | 41150 | 42334 |
| 710 | Osteomyelitis, periostitis, and other infections involving bone | musculoskeletal | 1.45 | 2.5E-23 | 1.46 | 2.1E-22 | 4169 | 165278 | 169447 | 1.33 | 3.1E-02 | 1222 | 41150 | 42372 |
| 710.19 | Unspecified osteomyelitis | musculoskeletal | 1.50 | 3.2E-20 | 1.51 | 5.2E-19 | 2917 | 165278 | 168195 | 1.46 | 1.8E-02 | 853 | 41150 | 42003 |
| 452 | Other venous embolism and thrombosis | circulatory system | 1.25 | 4.5E-21 | 1.25 | 3.5E-20 | 11454 | 126503 | 137957 | 1.20 | 3.9E-02 | 3106 | 32810 | 35916 |
| 452.2 | Deep vein thrombosis | circulatory system | 1.31 | 1.0E-17 | 1.32 | 2.2E-17 | 5966 | 126503 | 132469 | 1.19 | 1.3E-01 | 1781 | 32810 | 34591 |

with deep vein thrombosis (heterogeneity *p* >0.05 for both; **Fig 4**), two other circulatory system disorders associated at phenome-wide significance in the MR-PheWAS. A similar analysis examining associations of genetically-predicted height with several diabetes-related conditions stratified by diabetes status is described in online supplemental material (**S1 Text**).

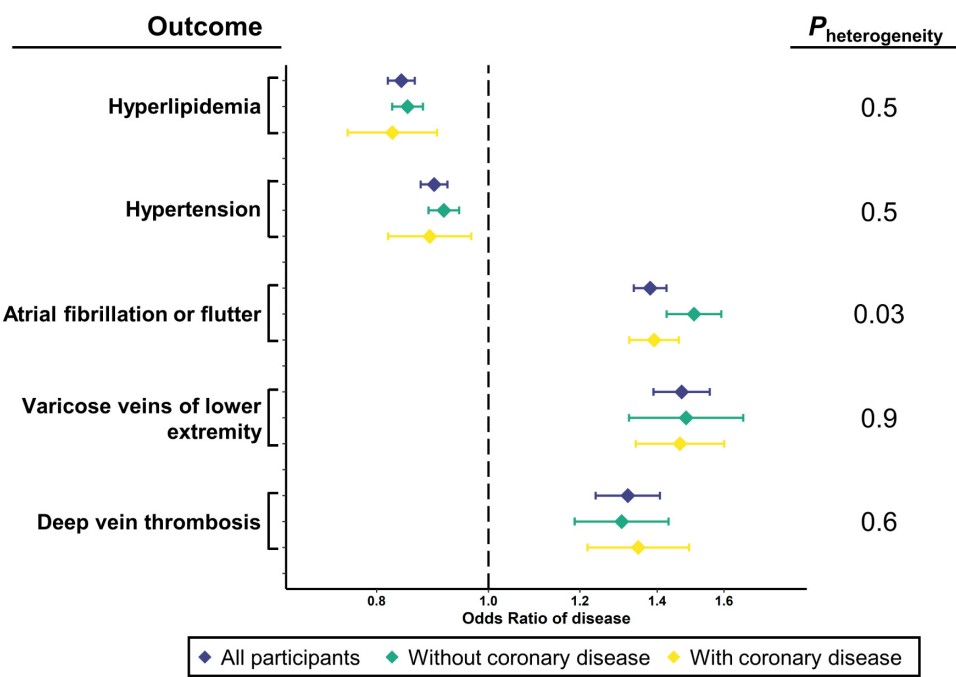

**Fig 4. Associations of selected traits with genetically-predicted height after stratifying by coronary heart disease status.** Odds ratio (OR) and 95% confidence intervals shown for associations of the indicated traits with genetically-predicted height in all participants (purple), those without coronary heart disease (green) and those with coronary heart disease (yellow). P-values from test of heterogeneity between strata shown to the right.

## Multi-population GWAS of height in the Million Veteran Program

The MR-PheWAS described above was performed using summary statistics from an external GWAS limited to individuals of European ancestry [10]. To determine if a multi-population GWAS in MVP might yield a substantial increase in the number of loci associated with height and thus inform a better instrument for estimating genetically-predicted height particularly in non-European ancestry samples in future studies, we performed GWAS in individuals of non-Hispanic White, non-Hispanic Black, and Hispanic-American race/ethnicity groups in MVP followed by multi-population meta-analysis. **Table 3** summarizes the GWAS results. GWAS in 235,398 EA individuals identified 18 height-associated loci with $p < 5 \times 10^{-8}$ that had not previously been reported in European-ancestry height GWAS [10] (**S8 Table**). Of 22 variants at these 18 loci, all 8 that were identified in summary statistics from Yengo et al [10] had a consistent direction of effect between MVP and the prior study (**S8 Table**). Three of the eight had p-values between $1 \times 10^{-8}$ and $5 \times 10^{-8}$ so were not reported in the prior study which employed a significance threshold of $p < 1 \times 10^{-8}$.

**Table 3. Summary of race/ethnicity-specific and multi-population meta-analysis genome-wide association study results.**

|  | Known loci | MVP EA | MVP AA | MVP HA | EA+AA+HA Multi-population meta-analysis |
|---|---|---|---|---|---|
| **Number of loci** | 712 | 533 | 20 | 16 | 488 |
| **Independent SNPs** | 3290 | 2,027 | 20 | 18 | 1,591 |
| **Novel loci** | - | 17 | 0 | 0 | 14 |
| **Independent SNPs at novel loci** | - | 21 | 0 | 0 | 21 |

We used LD score regression (LDSC) to examine heritability and the contribution of polygenicity to global inflation in the EA GWAS and to examine genetic correlation with the most recent European-ancestry GWAS meta-analysis [10,24,25]. We found SNP-based heritability of 0.40 (standard error [SE] of 0.02) in the MVP EA sample, an LDSC intercept of 1.15 (SE 0.04), and an attenuation ratio statistic of 0.07 (SE 0.02). The LDSC intercept in the MVP GWAS of EA individuals was lower than in the most recent European-ancestry GWAS [10]. As in the prior European-ancestry GWAS, the attenuation ratio statistic from LDSC was suggestive of polygenicity contributing to a large proportion of inflated test statistics rather than population stratification. Bivariate LDSC comparing MVP EA GWAS with summary statistics from a recent European-ancestry height GWAS found high genetic correlation of 0.98 (SE 0.007) suggesting very consistent genetic effects of height between MVP EA and prior European-ancestry GWAS.

GWAS in 63,898 AA individuals and 24,497 HA individuals did not identify any novel height-associated loci compared to published European-ancestry GWAS. Multi-population meta-analysis in MVP across all three race/ethnicity groups identified 16 loci represented by 24 variants associated with height not identified in prior height GWAS in European and African ancestry individuals [10,26] (**S9 Table**). Of the 24 variants at these 16 loci, only 1 was annotated as a non-synonymous coding variant, and none of the 24 had a scaled CADD score exceeding the 90[th] percentile, suggesting available genome annotation did not clearly indicate deleterious functional impact of the variants (**S10 Table**). We identified 6 of these 24 variants in European-ancestry GWAS summary statistics from Yengo et al [10] and 13 of the 24 variants in African-ancestry GWAS summary statistics from Graff et al [26]. All 6 identified in the prior European-ancestry GWAS summary statistics and all 13 identified in the prior African-ancestry GWAS summary statistics had concordant directions of effect with the multi-population GWAS results in MVP (**S9 Table**). Two of the six variants identified in prior European-ancestry GWAS summary statistics had genome-wide significant p-values in Yengo et al but were not among the 3290 variants retaining significance after the conditional and joint-multiple SNP analysis employed in that study. One of these loci–near the *PIP4K2A* gene–was also previously identified in a GWAS of height in Japanese individuals [27]. Taken together, potentially 14 loci, represented by 21 variants, did not have prior GWAS evidence of association with height in European- or African-ancestry individuals.

## Discussion

We report associations of genetically-predicted height with clinical traits across a spectrum of systems/domains. To our knowledge the broadest prior MR analysis of height examined 50 traits [15], and we expand that scope to over 1,000 traits using an MR-PheWAS approach applied to data from the largest integrated healthcare system in the US. We confirm known risk-increasing (atrial fibrillation/flutter) and risk-lowering (CHD, hypertension, hyperlipidemia) associations with cardiovascular conditions and risk factors, as well as recently reported associations with varicose veins [28,29]. In addition, we identified potentially novel associations with peripheral neuropathy and infections of the skin and bones. Although our sample had less statistical power to detect associations in AA individuals compared to EA individuals, we found generally consistent associations of genetically-predicted height with clinical conditions across the two populations. Meta-analysis of results from EA and AA individuals identified five additional traits associated with genetically-predicted height. Two were genitourinary conditions–erectile dysfunction and urinary retention–that can be associated with neuropathy, and a third was a phecode for non-specific skin disorders that may be related to skin infections–consistent with the race/ethnicity stratified results. The large sample available through

the MVP also permitted analyses stratified by CHD and diabetes mellitus status, revealing heterogeneity in the associations involving atrial fibrillation/flutter and infections common in diabetes patients, respectively. Finally, we found two traits associated with genetically-predicted height in women but not in men–asthma and non-specific peripheral nerve disorders. Whether these associations reflect differences by sex in disease pathophysiology related to height may warrant exploration in a sample with better balance between men and women. In sum, our results suggest that an individual's height may warrant consideration as a non-modifiable predictor for several common conditions, particularly those affecting peripheral/distal extremities that are most physically impacted by tall stature.

This PheWAS study confirms several associations of cardiovascular and circulatory system disorders with genetically-predicted height that have been previously described in other study samples but also provides some additional insights. First, the effect sizes for height associations with cardiovascular disease traits and risk factors were notably similar between the UK Biobank two-sample study and our one-sample two-stage least squares study in MVP, strengthening the weight of evidence supporting causally protective associations between tall stature and hypertension, hyperlipidemia, CHD, and atrial fibrillation. Second, the observation of a stronger association of genetically-predicted height with a lower risk of atrial fibrillation among individuals without CHD is consistent with negative confounding by CHD on the height-atrial fibrillation association. Lastly, we note a complete lack of association between genetically-predicted height and peripheral vascular disease (phecode 443). Whether this finding reflects truly divergent relationships of height with disease in distinct vascular beds warrants future study. The PheWAS results also extend our understanding of the clinical impacts of tall stature beyond cardiovascular disease. Notably, increased stature was disease risk-increasing for the majority of non-cardiovascular conditions, contrary to the pattern of association with cardiovascular disease risk factors and CHD. Two clusters of traits of particular interest were peripheral neuropathy conditions and venous circulatory disorders.

Studies examining risk factors for slowed nerve conduction have previously found an inverse association of height with nerve conduction velocity and amplitude [30,31]. The association of genetically-predicted height with clinical peripheral neuropathy supports the prior epidemiologic findings and suggests that height-related effects on nerve conduction are clinically significant. We also observed associations of genetically-predicted height with extremity complications that are not infrequently observed in the setting of peripheral neuropathy including cellulitis and skin abscesses, chronic leg ulcers, and osteomyelitis. Recent work has found height and peripheral sensory neuropathy to be independent risk factors for diabetic foot ulcer [32,33]. We found consistent associations of genetically-predicted height with peripheral neuropathy and chronic leg ulcer irrespective of diabetes status. In contrast, we observed stronger associations of genetically-predicted height with skin and bone infections in those with diabetes compared to those without diabetes, suggesting synergy between taller stature and other characteristics of diabetes and diabetes care to impact infection risk. To our knowledge, height has not been described as a risk factor for skin and bone infections, in those with or without diabetes, though a plausible mechanism would be via height-related peripheral neuropathy.

Prior observational studies have suggested increased height predisposes to varicose veins and the causality of these associations has been supported through MR analyses [29]. Buttressing these epidemiologic observations are studies demonstrating adverse venous pressure dynamics in taller individuals that likely promote peripheral venous stasis and varicose veins [34]. These findings were further supported by a second genetic study of varicose veins that found standing height and height at age 10 were causally associated with varicose veins in MR analyses [28]. In this PheWAS of genetically-predicted height in MVP, we found evidence

supporting potentially causal associations of height with varicose veins and venous thrombo-embolic events and extend that association to a number of other related venous circulatory disorders: chronic venous insufficiency and venous hypertension.

Our multi-population GWAS in MVP identified 14 height-associated loci that were not found in recent European-ancestry or African-ancestry GWAS meta-analysis. Interestingly, at least three of the loci identified in the multi-population meta-analysis fall near genes–*HMCN1*, *DLG5*, and *SMURF2* –that were not found in conventional GWAS of height in European-ancestry individuals but were identified in analyses that incorporated functional annotation into the association analysis [35]. Another locus, near the *BMP2K* gene, is highly plausible as a height-associated locus given that its expression is inducible by *BMP2*, a gene that was also associated with height in the aforementioned analysis [35]. The *USP44* gene, harboring the only non-synonymous variant identified in this multi-population GWAS meta-analysis, has been associated with type 2 diabetes [36,37], C-reactive protein levels [38], and acute myeloid leukemia [39] in prior GWAS, but has not been implicated in anthropometric traits in prior studies. The multi-population meta-analysis in MVP also replicated one locus on chromosome 10 (near the *ECD* gene) that was identified in a recent African-ancestry GWAS of height [26] and that had not previously been discovered in European-ancestry GWAS. The identification of additional genetic associations with height in a smaller total sample size than the most recent European-ancestry GWAS supports the importance of non-European populations in characterizing the genetic architecture of complex traits as has been well-described previously [40,41]. Indeed, a multi-cohort, international multi-ancestry height GWAS meta-analysis is nearly complete and will extend the single-study results we report here [42].

Important limitations to our analyses exist. First, we used loci associated with height from a European-ancestry GWAS meta-analysis to develop the GRS employed in the MR-PheWAS analysis. Thus, the analysis of non-Hispanic Black individuals was limited by a weaker genetic instrument. As multi-ancestry height GWAS meta-analysis are completed, stronger genetic instruments for MR-PheWAS analyses in AA and other non-EA populations may soon be available. Second, we had a substantial discrepancy in sample size between EA and AA individuals in MVP which undoubtedly contributed to the variation in the number of traits associated with genetically-predicted height at phenome-wide significance between the two race/ethnicity groups. As has been recognized for GWAS, only with expansion of non-European ancestry and non-White race/ethnicity samples will we be able to determine if phenome-wide associations of traits with genetically-predicted height are common across or vary between ancestry and race/ethnicity groups. Third, we did not interrogate genetic correlation or pleiotropy between height and associated traits identified in the MR-PheWAS. To perform such analyses on a phenome-wide scale exceeds the scope of this manuscript. Thus, we are cautious about causal interpretation of the results of the MR-PheWAS reported here in the absence of such secondary analyses. Given increasing numbers of clinical biobanks globally, replication of MR-PheWAS associations in independent cohorts will be the focus of future work. Fourth, prior work has demonstrated associations of genetically-estimated height with income and socioeconomic status particularly in men [6], and we cannot exclude the possibility that associations found in the MR-PheWAS are mediated by socioeconomic status rather than a direct effect of height. Income and education data is available in only a subset MVP of participants, limiting the ability to comprehensively evaluate mediation by socioeconomic variables at the present time. Finally, the sample of individuals receiving care in the US VA Healthcare System may not represent a general US adult population. In particular, US Veterans in this study are mostly older males with higher prevalence of a number of common chronic conditions, including diabetes and cardiovascular disease. While the higher burden of disease may make the MVP sample non-representative of a typical adult population, the higher prevalence of

many traits enhances statistical power for detecting associations in the PheWAS and MR-PheWAS.

In conclusion, we found genetic evidence supporting associations between height and 127 EHR traits in individuals of non-Hispanic White individuals, 48 of which exhibited nominally-significant associations with genetically-predicted height in non-Hispanic Black individuals. While much work has focused on inverse associations of genetically-predicted height with CHD and its risk factors, this MR-PheWAS analysis suggests taller stature is associated with higher prevalence of many other clinically relevant traits. In particular, we describe associations of genetically-predicted height with conditions that may result from the effects of increased weight-bearing such as acquired toe and foot deformities, and with peripheral neuropathy traits and venous circulatory disorders, conditions for which epidemiologic and physiologic studies have previously suggested a height-dependence. Finally, we highlight the potential importance of height as a risk factor that can impact the care of common chronic diseases by demonstrating interactions of height with diabetes mellitus on skin and bone infections. Taken together, we conclude that height may be an under recognized non-modifiable risk factor for a wide variety of common clinical conditions that may have implications for risk stratification and disease surveillance.

## Methods

### Ethics statement

Ethical oversight and human subjects research protocol approval for the MVP were provided by the VA Central Institutional Review Board (approval number 16–06) in accordance with the principles outlined in the Declaration of Helsinki.

### Study participants

The design of the MVP has been previously described [43,44]. Briefly, participants were recruited from over 60 VA medical centers nationwide starting in 2011 and provided written informed consent for participation. Enrolled Veterans provided a blood sample for banking from which DNA was extracted for genotyping. Phenotype data for the MVP is derived from the VA electronic health record (EHR), integrating inpatient and outpatient International Classification of Diseases (ICD 9/10) diagnosis and procedure codes, Current Procedural Terminology (CPT) procedure codes, clinical laboratory measurements, medications, and reports of diagnostic imaging modalities into a clinical research database.

### Anthropometric traits

Height and body mass index (BMI) were based on EHR data from clinical examinations available for participants from 2003 through 2018. For height, we used the average of all measurements available for a participant, excluding height measurements that were >3 inches above or below the average for each participant. Individuals with extreme average heights (≤50 inches or ≥100 inches) were also excluded. Average height was then converted to centimeters for all subsequent analyses. BMI was calculated as the weight (in kilograms) divided by the height (in meters) squared. We calculated the average BMI using all measurements occurring within 1.5 years before to 1.5 years after the date of MVP enrollment excluding weight measurements >60 pounds from the average of each participant.

### Genetic data

Genotyping, quality control procedures, and imputation in the MVP has been described previously [43]. Briefly, DNA extracted from blood was genotyped using a customized Affymetrix

Axiom biobank array (MVP 1.0 Genotyping Array) with content enriched for common and rare genetic variants of clinical significance. Duplicate samples and samples with excess heterozygosity, with >2.5% of missing genotype calls, or with sex-gender discordance were excluded. In cases of pairs of related individuals, one individual from each pair was removed. Variants with poor calling or with allele frequencies discrepant from 1000 Genomes Project [45] reference data were excluded. Genotypes from the 1000 Genomes Project phase 3 reference panel were imputed into MVP as described previously [46]. After imputation, variant level quality control was performed with the following exclusion thresholds: ancestry-specific Hardy-Weinberg equilibrium [47] $p$-value $<1x10^{-20}$, posterior call probability $< 0.9$, imputation quality $<0.3$, minor allele frequency (MAF) $< 0.0003$, call rate $< 97.5\%$ for common variants (MAF $> 1\%$), and call rate $< 99\%$ for rare variants (MAF $< 1\%$). Finally, we performed global and ancestry-specific principal component analysis (PCA) using the flashPCA software.

### Genetic risk score for height

We used 3,290 independent, genome-wide significant variants and their beta coefficients from a previously published GWAS in individuals of European ancestry [10] with no overlap with MVP to build a genetic risk score (GRS) for height. The weighted genetic risk score for height was the sum of the number of height-raising alleles in each individual, weighting each variant by the beta estimate for its effect on height from the published source GWAS [10].

### PheWAS phenotypes

Case/control status of MVP participants for 1,813 phecodes [48,49] was assigned using EHR data, based on data available at the time of enrollment into the MVP, and required at least two qualifying diagnosis codes to be classified as a case for a phecode. Analyses were limited to phecodes with at least 200 case and 200 non-case participants.

### PheWAS and MR-PheWAS of height

Race/ethnicity group was assigned using the Harmonized Ancestry and Race/Ethnicity (HARE) algorithm, which integrates self-identified race/ethnicity with genetically inferred ancestry [50]. As the HARE algorithm uses genetic data to predict race/ethnicity [50], we describe the populations studied here as race/ethnicity groups rather than ancestry groups and describe the study as a multi-population analysis rather than multi-ancestry. Accordingly, we classified individuals as non-Hispanic White (EA), non-Hispanic Black (AA), and Hispanic-American (HA) using HARE. PheWAS analyses were performed in a total of 222,300 EA and 58,151 AA participants with non-missing measured height, height GRS, and EHR phenotype data. To estimate associations of height with EHR traits, we performed race/ethnicity-stratified logistic regression of phecodes as the dependent variable and height Z-scores as the independent variable, adjusting for age at MVP enrollment, sex, and 10 race/ethnicity-specific PCs [50,51].

In the MR-PheWAS, associations of EHR traits with genetically-estimated height were performed using a two-stage least squares regression [13,21]. In the first stage, we performed race/ethnicity-stratified linear regression of measured height (dependent variable) on the height GRS, adjusting for 10 race/ethnicity-specific PCs [50,51]. Next, we estimated genetically-predicted height for each participant using the first-stage regression equation, and standardized the resulting genetically-predicted height for each participant to the standard deviation of measured height. Thus, a one-unit change in the standardized genetically-predicted height corresponds to a one-unit change in the height Z-score. In the second stage, we used the R *PheWAS* package [52] to perform a race/ethnicity-stratified MR-PheWAS. This package uses

logistic regression to estimate associations of phecodes as the dependent variables and the standardized genetically-predicted height estimated from the first stage as the independent variable, adjusting for age at enrollment in MVP, sex, and 10 race/ethnicity-specific PCs. Results of the MR-PheWAS, therefore, indicate the odds ratio (OR) for a phecode per unit change in genetically-predicted height, scaled to the standard deviation of measured height in our sample to allow direct comparability between the PheWAS and MR-PheWAS results. The number of phecodes included in the analysis of EA individuals was 1,378 and of AA individuals was 997, resulting in corrected thresholds for significance of $p<3.6\times10^{-5}$ (0.05 / 1,378) for EA and $p<5.0\times10^{-5}$ (0.05 / 997) for AA analyses. We reported nominally-significant association ($p<0.05$) results of the PheWAS in AA given the much smaller sample size and the fact that the GRS weights were derived from a European-ancestry GWAS. A significant association in this second-stage of the MR-PheWAS suggests an association of the phecode with variation in height due to genetic variants not merely an association of the phecode with measured height or with the GRS.

We tested concordance in direction of effect of phenome-wide significant associations of traits with measured height and with genetically-predicted height associations between EA and AA analyses using chi-square tests. For each of these pairwise comparisons, we also standardized effect estimates (beta coefficients) to the standard error of the effect estimate and the square root of the sample size within race/ethnicity group to create a standardized Z' effect size for each race/ethnicity group [53,54]. We then compared the standardized Z' effect estimates for phenome-wide significant associations (with measured height and with genetically-predicted height) from EA and AA using the slope of the linear regression of beta coefficients arising from each pair of analyses being compared (e.g., height-trait associations between EA and AA). Finally, we performed a fixed effects meta-analysis of MR-PheWAS results with genetically-predicted across EA and AA race/ethnicity groups in METAL [55], applying a significance threshold of $p<3.6\times10^{-5}$ corresponding to the threshold for the race/ethnicity group with the larger sample size (EA).

## Secondary analyses

We performed several secondary analyses motivated by the results of the PheWAS and previously known relationships between height and clinical characteristics. As height and body mass index (BMI; calculated as weight in kilograms divided by height in meters squared) are correlated, we ran an additional model with BMI as an added covariate to determine the extent to which genetically-predicted height associations with phecodes could be confounded by BMI. In addition, we repeated the PheWAS stratified by diabetes and CHD status to address two potential issues. First, differential recognition and documentation in the EHR of conditions in individuals with diabetes or CHD due to more frequent or intense clinical care could induce false positive associations in the MR-PheWAS. This is particularly of concern for CHD-associated conditions given the previously described associations of height with CHD. Second, the stratified analyses might reveal differential associations of genetically-predicted height with conditions that could suggest synergistic effects of diabetes or CHD with height on those conditions. As some of the traits most strongly associated with genetically-predicted height are potentially complications of or correlated with diabetes or CHD, we selected these common conditions that are more highly prevalent among Veterans than in the general population [56–58] for the stratified analyses. For diabetes, stratification was based on the presence/absence of phecodes related to diabetes and diabetes-related conditions: 250, 250.1, 250.11, 250.12, 250.13, 250.14, 250.15, 250.2, 250.21, 250.22, 250.23, 250.24, 250.25, 250.6, and 249. For CHD, stratification was based on the presence/absence of phecodes related to coronary atherosclerosis: 411, 411.1, 411.2, 411.3, 411.4, 411.41, 411.8, and 411.9.

## GWAS

Lastly, we performed a GWAS of height in the MVP sample for two reasons. First, we wanted to compare the genetic associations with height in MVP of the variants used to generate the height GRS across non-European ancestries. Second, we wanted to leverage the multi-population MVP sample containing a substantial sample size of non-EA participants to perform a multi-population meta-analysis of height to determine if inclusion of diverse ancestries yielded novel genetic associations with height. GWAS of height in the MVP cohort was examined separately among EA (N = 235,398), AA (N = 63,898), and HA (N = 24,497) participants based on the HARE algorithm for classifying race/ethnicity[50]. For each race/ethnicity group, height was stratified by sex and adjusted for age, age$^2$, and the top ten genotype-derived PCs in a linear regression model. The resulting residuals were transformed to approximate normality using inverse normal scores. Imputed and directly measured genetic variants were tested for association with the inverse normal transformed residuals of height through linear regression assuming an additive genetic model using PLINK2[59]. We performed inverse-variance weighted fixed-effects meta-analysis using METAL [55]. For the minority ancestries in our sample, we meta-analyzed the GWAS results from MVP AA and HA participants, and we performed a multi-population meta-analysis within MVP of EA, AA, and HA results. GWAS results were summarized using FUMA (http://fuma.ctglab.nl/), a platform that annotates, prioritizes, visualizes and interprets GWAS results [60]. Independent, genome-wide significant SNPs were defined as those with $p < 5 \times 10^{-8}$ and with LD $r^2 < 0.6$ with each other. SNPs with $p < 0.05$ were grouped into a genomic locus if they were linked at $r^2 \geq 0.6$ or were physically close (distance < 500kb). Lead SNPs were defined within each locus if they were independent ($r^2 < 0.1$) and genome-wide significant ($p < 5 \times 10^{-8}$). Novel loci were defined as those with genome-wide significance ($p < 5 \times 10^{-8}$) and a distance > 500kb from previously published variants in European and African ancestries [10,26]. We performed linkage disequilibrium score regression (LDSC) to quantify heritability, population stratification in the MVP EA GWAS and to examine genetic correlation between GWAS in MVP EA individuals and prior European-ancestry GWAS meta-analysis [24,25]. We used LDSC software available at https://github.com/bulik/ldsc and LD scores for 1000 genomes project European-ancestry individuals available with the software [24,25].

To quantify the transferability of the beta coefficients from the European-ancestry GWAS used to build the height GRS to the MVP sample, we tested directional concordance of the association with height of the 3,290 variants included in the GRS using a chi-square test. Additionally, we calculated the standardized Z'[53,54] for the effect size of variant associations with height in MVP and in the published GWAS and estimated the slope of the regression line between effect sizes in the two samples with a slope of 1 in the regression indicating equivalent effect sizes in both samples.

## Supporting information

**S1 Text. Effect of diabetes mellitus status on genetically-predicted height associations with neurologic, dermatologic, and infectious diabetes complications. Fig A.** Associations of selected traits with genetically-predicted height after stratifying by diabetes status. Odds ratio (OR) and 95% confidence intervals shown for associations of the indicated traits with genetically-predicted height in all participants (purple), those without diabetes (green), and those with diabetes (yellow). P-values from test of heterogeneity between strata shown to the right. (PDF)

**S1 Fig.** Comparison of standardized effect sizes (Z') for associations with height of variants used in height genetic risk score for MR-PheWAS in Yengo et al (*Human Molecular Genetics*

2018:27(20):3641–3649) and MVP non-Hispanic White (EA, left) and non-Hispanic Black (AA, right) individuals.
(PDF)

**S2 Fig.** Effect size comparison of associations of genetically-predicted height with clinical traits in MR-PheWAS without and with body mass index (BMI) as a covariate in non-Hispanic White (A) and non-Hispanic Black (B) individuals.
(PDF)

**S1 Table. Phenome-wide significant trait associations with measured height in non-Hispanic White individuals.**
(XLSX)

**S2 Table. Phenome-wide significant trait associations with measured height in non-Hispanic Black individuals.**
(XLSX)

**S3 Table. Phenome-wide significant trait associations with genetically-predicted height in non-Hispanic White individuals.**
(XLSX)

**S4 Table. Phenome-wide significant traits from non-Hispanic White individuals with at least nominally significant associations with genetically-predicted height in non-Hispanic Black individuals.**
(XLSX)

**S5 Table. Comparison of 11 traits associated with genetically-predicted height in UK Biobank with analogous phecodes in MVP.**
(XLSX)

**S6 Table. Traits associated with genetically-predicted height at phenome-wide significance only after multi-population meta-analysis.**
(XLSX)

**S7 Table. Sex-stratified results for six traits associated with genetically-predicted height at phenome-wide significance in non-Hispanic White women.**
(XLSX)

**S8 Table. Novel loci associated at genome-wide significance in non-Hispanic White individuals.**
(XLSX)

**S9 Table. Novel loci associated at genome-wide significance in trans-ethnic meta-analysis of non-Hispanic White, non-Hispanic Black, and Hispanic-American individuals.**
(XLSX)

**S10 Table. Genomic annotation and regulatory function CADD scores for height-associated loci not identified in published European-ancestry GWAS.**
(XLSX)

**S11 Table. Phenome-wide association study results for measured height in non-Hispanic White individuals in MVP.**
(XLSX)

**S12 Table. Phenome-wide association study results for measured height in non-Hispanic Black individuals in MVP.**
(XLSX)

**S13 Table. Phenome-wide association study results for genetically-predicted height in non-Hispanic White individuals in MVP.**
(XLSX)

**S14 Table. Phenome-wide association study results for genetically-predicted height in non-Hispanic Black individuals in MVP.**
(XLSX)

## Acknowledgments

This publication does not represent the views of the Department of Veteran Affairs or the United States Government. We are grateful to our Veterans for their contributions to MVP.

## Author Contributions

**Conceptualization:** Sridharan Raghavan, Jie Huang, Catherine Tcheandjieu, Kari E. North, Benjamin F. Voight, Christopher J. O'Donnell, Yan V. Sun, Themistocles L. Assimes.

**Data curation:** Jennifer E. Huffman, Elizabeth Litkowski, Yuk-Lam A. Ho, Haley Hunter-Zinck, J. Michael Gaziano, Saiju Pyarajan, Kelly Cho, Christopher J. O'Donnell, Yan V. Sun, Themistocles L. Assimes.

**Formal analysis:** Sridharan Raghavan, Jie Huang, Catherine Tcheandjieu, Elizabeth Litkowski, Chang Liu.

**Funding acquisition:** Sridharan Raghavan, Philip S. Tsao, Peter W. F. Wilson, Kyong-Mi Chang, Kelly Cho.

**Investigation:** Sridharan Raghavan, Jie Huang, Catherine Tcheandjieu, Jennifer E. Huffman, Elizabeth Litkowski, Chang Liu.

**Methodology:** Jie Huang, Catherine Tcheandjieu, Jennifer E. Huffman, Haley Hunter-Zinck, Hongyu Zhao, Eirini Marouli, Kari E. North, Ethan Lange, Leslie A. Lange, Benjamin F. Voight, Christopher J. O'Donnell, Yan V. Sun, Themistocles L. Assimes.

**Project administration:** J. Michael Gaziano, Philip S. Tsao, Peter W. F. Wilson, Kyong-Mi Chang, Kelly Cho, Christopher J. O'Donnell.

**Resources:** Sridharan Raghavan, Hongyu Zhao, J. Michael Gaziano, Saiju Pyarajan, Elizabeth R. Hauser, Philip S. Tsao, Peter W. F. Wilson, Kyong-Mi Chang, Kelly Cho, Christopher J. O'Donnell, Yan V. Sun, Themistocles L. Assimes.

**Software:** Saiju Pyarajan, Elizabeth R. Hauser, Yan V. Sun.

**Supervision:** Ethan Lange, Leslie A. Lange, J. Michael Gaziano, Philip S. Tsao, Peter W. F. Wilson, Kyong-Mi Chang, Kelly Cho, Christopher J. O'Donnell, Yan V. Sun, Themistocles L. Assimes.

**Visualization:** Sridharan Raghavan, Jie Huang, Catherine Tcheandjieu.

**Writing – original draft:** Sridharan Raghavan, Jie Huang, Catherine Tcheandjieu, Jennifer E. Huffman, Christopher J. O'Donnell, Yan V. Sun, Themistocles L. Assimes.

**Writing – review & editing:** Sridharan Raghavan, Jie Huang, Catherine Tcheandjieu, Jennifer E. Huffman, Elizabeth Litkowski, Yuk-Lam A. Ho, Haley Hunter-Zinck, Hongyu Zhao, Eirini Marouli, Kari E. North, Ethan Lange, Leslie A. Lange, Benjamin F. Voight, J. Michael Gaziano, Saiju Pyarajan, Elizabeth R. Hauser, Philip S. Tsao, Peter W. F. Wilson, Kyong-Mi Chang, Kelly Cho, Christopher J. O'Donnell, Yan V. Sun, Themistocles L. Assimes.

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
