## [Decision Letter · Decision Letter 0]

8 Dec 2021

Dear Dr Raghavan,

Thank you very much for submitting your Research Article entitled 'A multi-ancestry phenome-wide association study of genetically-predicted height in the Million Veteran Program' to PLOS Genetics.

The manuscript was fully evaluated at the editorial level and by independent peer reviewers. The reviewers appreciated the attention to an important problem, but raised some substantial concerns about the current manuscript. Based on the reviews, we will not be able to accept this version of the manuscript, but we would be willing to review a much-revised version. We cannot, of course, promise publication at that time.

If you decide to revise the manuscript for further consideration at PLOS Genetics, please aim to resubmit within the next 60 days, unless it will take extra time to address the concerns of the reviewers, in which case we would appreciate an expected resubmission date by email to plosgenetics@plos.org.

[LINK]

We are sorry that we cannot be more positive about your manuscript at this stage. Please do not hesitate to contact us if you have any concerns or questions.

Yours sincerely,

Greg Gibson

Consulting Editor - PLoS Genetics

PLOS Genetics

Scott Williams

Section Editor: Natural Variation

PLOS Genetics

We have received detailed responses from two expert reviewers, both of which provide thoughtful suggestions for improvement upon revision. These edits will both clarify some outstanding issues and add clarity with some new analyses. Overall, the appraisal of the manuscript is very positive.

Reviewer's Responses to Questions

**Comments to the Authors:**

Reviewer #1: In this study, Raghavan et al. quantify the effect of a polygenic predictor of height on various clinical conditions in a large ancestrally diverse sample from the Million Veteran Program. Overall, they find hundreds of associations between height and various disease, including previously reported signals. This study is an important contribution to the literature, which demonstrates that height is indeed an underappreciated risk factor for many diseases. However, I found that the study did not fully utilise the potential of the dataset and raise below a points that could improve the quality of the study.

Major points (in no particular order)

1. I found the GRS analysis to be the most interesting part of the manuscript. The phenotypic analysis (height vs disease) could have been utilised as a way filter diseases to focus on but the fact that the authors present those analyses in parallel, reduces the interest of the phenotypic analysis alone.

2. The authors report what seems to be a good concordance of the effect of GRS on disease risk across ancestries. Such an observation calls for a meta-analysis across ancestry groups, which may also increase power.

3. The authors perform a GWAS in a European ancestry sample of the MVP. This analysis is interesting but they authors do not use those results to further improve power to address their main question: the association between height PRS and disease. By meta-analysing Yengo et al. with their MVP-EA GWAS, they could have increased power to detect association in African American.

4. The authors rightly underline that an association between height GRS and disease may not necessarily reflect causality but possibly pleiotropy. Nevertheless, as I reader I’d like to know how many (statistically) independent associations are reported here. The grouping in terms of broad disease category partly addresses my question but it still possible that pleiotropic effect exist beyond phecodes. For example, is the effect of GRS on Atrial fibrillation independent of that on varicose veins?

Minor points

Line 153 (and elsewhere). Please specify what’s X and Y when reporting regression slopes. Do you account for estimation errors in both samples when estimating those slopes? If you then the authors could use the method developed by Qi et al. Nat Comm. 2018 (PMID 29891976)

Line 161 – 164. Note that these comparisons are affected by winner’s curse. So I’d suggest correcting estimates for that before estimating those slopes.

Line 186. “BMI as a coefficient” => “BMI as a covariate”

Line 193 (and elsewhere). Specify the unit for the that OR. Is that risk increase per height SD or per GRS SD? I’d assume the former given that the authors use a two-stage least squares approach. Could you please clarify that?

The diabetes analysis looks a bit artificial given that there is no statistically significant overlap between diabetes related complications and disease associated with height. The authors write (lines 179) “we noted that a number of conditions in Table 2 are associated with diabetes…”, which is no rigorous enough. So unless they can prove that there is such an enrichment, I’d suggest removing all diabetes-related sections from the main text to improve clarity.

Line 308. It seems a bit odd that there could be a bi-directional causal effect between varicose veins (VV) and height given that VV is classically diagnosed in >18y. Can they authors revise that sentence (e.g., by commenting on age of onset)?

Line 313. I think the authors should comment on why the 16 height-associated variants identified in their GWAS have not been detected before? Can they compare effect sizes and other GWAS? For example, that of Graff et al. AJHG 2021 (PMID 33713608)

Sorry if I missed it but the authors should specify what software they have used for running GWAS. PLINK? BOLT-LMM? Also, can the authors report results from LD score regression univariate (in MVP alone) and bivariate to compare with Yengo et al.?

Loic Yengo

Reviewer #2: This manuscript by Raghavan et al. describes an MR-PheWAS looking for associations between height and other EHR traits. The authors use the VA Million Veterans Program (MVP) for the study, which is one of the largest datasets available with EHR data from diverse ancestry groups. Overall, this is a very good study with some interesting results.

There are a few areas of the manuscript that would benefit from additional information.

Major revisions

- In line 103-105, the authors indicate that a similar study was performed in 50 traits from UK Biobank. Do any of the results from this paper replicate in UK Biobank? Were the 50 traits included in that published work included here? It does not look like the authors attempted to replicate the findings from this paper in the UK Biobank, but that seems like an obvious next step and may greatly improve the paper.

- Since the VA MVP participants are 91.6% male, did the authors consider doing a sex-stratified analysis? I am a little concerned that if there is any trait that has an unusual case distribution by sex, then the results could be confounded especially due to the drastic skew in the dataset.

- On lines 130-134, the authors describe how the sample sizes of the study participants included. Did you leave out HA because of sample size? I think it is important to state that. Also, why is the number of EA and AA lower than the first line of the paragraph? Why did people drop off?

- On line 139, the variance explained for height in the AA population is pretty low. Did you also try a meta-analysis from AA? Perhaps from this study: https://www.sciencedirect.com/science/article/pii/S0002929721000537#mmc2

- On line 155-156, the authors indicate that the phewas associations were directionally concordant. If they are concordant across ancestry then how do you have EA only and AA only results listed? This is not clear. Also on lines 164-166, how is it 142 traits concordant across ancestry when only 2 were significant in AA (46 more nominally)? This is confusing.

-On line 170, the authors indicate that 2 traits are significant in AA. The number of significant traits is substantially different between ancestry groups. Any idea what is driving this? Is it power and sample size? Is it differences in health care utilization with different volumes of EHR data and diagnoses? Is it because the AA group is much younger so less diagnoses? Or is it biology?

- The relationship between diabetes and height is not totally clear at the end of the section ending on line 212. Perhaps this section needs a summary at the end.

-On line 420, did case status apply to individuals with one instance of the ICD code or did you require a rule of 2 or 3?

- On line 436, How did you decide on 10 PCs? How much variance was explained by these?

Minor revisions

- Line 128, the abbreviation "HA" was not yet defined.

- Line 137, the abbreviation "GRS" was not yet defined.

- Throughout the paper, the authors use race/ethnicity to define the groups, however using HARE, they are also using ancestry. I wonder if the word "ancestry" should be used? Or perhaps race/ethnicity/ancestry? They mean different things of course. But since genetics was used to define, I think it is important to include ancestry in the paper. Similarly, should it be "trans-ancestry" or "cross-ancestry"?

- On line 198, there is something odd with reference.

- What software was used for the GWAS?

**Have all data underlying the figures and results presented in the manuscript been provided?**

Reviewer #1: Yes

Reviewer #2: Yes

PLOS authors have the option to publish the peer review history of their article (what does this mean?). If published, this will include your full peer review and any attached files.

Reviewer #1: **Yes: **Loic Yengo

Reviewer #2: **Yes: **Marylyn D Ritchie

---

## [Decision Letter · Decision Letter 1]

5 Apr 2022

Dear Dr Raghavan,

We are pleased to inform you that your manuscript entitled "A multi-population phenome-wide association study of genetically-predicted height in the Million Veteran Program" has been editorially accepted for publication in PLOS Genetics. Congratulations!

Yours sincerely,

Greg Gibson

Consulting Editor - PLoS Genetics

PLOS Genetics

Scott Williams

Section Editor: Human Variation

PLOS Genetics

Comments from the editors:

Thank you for your thorough response. One of the initial reviewers was available to provide his/her second opinion, and the editors have also considered the response. We are well satisfied.

Reviewer's Responses to Questions

**Comments to the Authors:**

Reviewer #1: The authors have convincingly addressed all my comments.

**Have all data underlying the figures and results presented in the manuscript been provided?**

Reviewer #1: Yes

PLOS authors have the option to publish the peer review history of their article (what does this mean?). If published, this will include your full peer review and any attached files.

Reviewer #1: No

**Data Deposition**

http://datadryad.org/submit?journalID=pgenetics&manu=PGENETICS-D-21-01241R1

**Press Queries**

---

## [Editor Report · Acceptance letter]

9 May 2022

PGENETICS-D-21-01241R1 

A multi-population phenome-wide association study of genetically-predicted height in the Million Veteran Program 

Dear Dr Raghavan, 

We are pleased to inform you that your manuscript entitled "A multi-population phenome-wide association study of genetically-predicted height in the Million Veteran Program" has been formally accepted for publication in PLOS Genetics! Your manuscript is now with our production department and you will be notified of the publication date in due course.

With kind regards,

Livia Horvath

PLOS Genetics

On behalf of:
